# Receptor-Independent Anti-Ferroptotic Activity of TrkB Modulators

**DOI:** 10.3390/ijms232416205

**Published:** 2022-12-19

**Authors:** Md. Jakaria, Abdel A. Belaidi, Adam Southon, Krista A. Dent, Darius J. R. Lane, Ashley I. Bush, Scott Ayton

**Affiliations:** The Florey Institute of Neuroscience and Mental Health, The University of Melbourne, Parkville, VIC 3052, Australia

**Keywords:** BDNF, TrkB, agonist, antagonist, inhibitor, protection, ferroptosis

## Abstract

Dysregulated brain-derived neurotrophic factor (BDNF)/tropomyosin receptor kinase B (TrkB) signalling is implicated in several neurodegenerative diseases, including Alzheimer’s disease. A failure of neurotrophic support may participate in neurodegenerative mechanisms, such as ferroptosis, which has likewise been implicated in this disease class. The current study investigated whether modulators of TrkB signalling affect ferroptosis. Cell viability, C11 BODIPY, and cell-free oxidation assays were used to observe the impact of TrkB modulators, and an immunoblot assay was used to detect TrkB expression. TrkB modulators such as agonist BDNF, antagonist ANA-12, and inhibitor K252a did not affect RSL3-induced ferroptosis sensitivity in primary cortical neurons expressing detectable TrkB receptors. Several other modulators of the TrkB receptor, including agonist 7,8-DHF, activator phenelzine sulphate, and inhibitor GNF-5837, conferred protection against a range of ferroptosis inducers in several immortalised neuronal and non-neuronal cell lines, such as N27 and HT-1080 cells. We found these immortalised cell lines lack detectable TrkB receptor expression, so the anti-ferroptotic activity of these TrkB modulators was most likely due to their inherent radical-trapping antioxidant properties, which should be considered when interpreting their experimental findings. These modulators or their variants could be potential anti-ferroptotic therapeutics for various diseases.

## 1. Introduction

Tropomyosin receptor kinase B (TrkB; gene: *NTRK2*; tropomyosin receptor kinase family) is a receptor for brain-derived neurotrophic factor (BDNF) and neurotrophin 4. The binding of ligands to the full-length TrkB receptor leads to ligand–receptor dimerisation and autophosphorylation of tyrosine residues in the intracellular kinase domain of the receptor [1,2,3]. This, in turn, activates an intracellular downstream signalling cascade crucial for regulating cell proliferation, survival and death, neurite outgrowth, neuronal transmission, synaptogenesis, and synaptic plasticity. In contrast, dysregulation of this signalling is implicated in several neurological disorders, cancers, obesity, and eating disorders [2].

Evidence for aberrant TrkB signalling has been found in several neurodegenerative diseases, including Alzheimer’s disease (AD) [4,5,6]. BDNF and full-length TrkB levels are reported as decreased, while TrkB-truncated levels are increased in hippocampal and cortical post-mortem AD tissue [7]. The BDNF Val66Met (rs6265) polymorphism has been associated with a steeper decline in episodic memory and executive function as well as a greater Aβ burden than the BDNF Val allele [8,9].

Ferroptosis, a unique form of iron-mediated regulated cell death, is executed by the build-up of lethal lipid hydroperoxides [10,11,12]. In experimental models, ferroptosis can be induced by numerous chemical agents such as erastin, l-buthionine-sulfoximine (BSO), RSL3, ammonium iron (II) sulphate, and peroxide compounds such as an endoperoxide-containing 1,2-dioxolane (FINO2) and tert-butyl hydroperoxide (tBH) [11,13]. The induced ferroptosis can be aborted by lipophilic radical-trapping agents, such as liproxstatin 1 (Lip-1) and diacetylbis (N(4)-methylthiosemicarbazonato) copper (II) (Cu^II^(ATSM)), and by iron chelators such as deferiprone (DFP) [11,14]. Ferroptosis has been implicated in the progression of several neurodegenerative conditions [15]. Therefore, finding favourable anti-ferroptotic molecules is of interest as candidate treatments for neurodegenerative diseases such as AD.

Growing evidence suggests that the ligand-activated TrkB signalling pathway mediates neuroprotection via amelioration of aberrant cellular responses such as inflammation [16,17], oxidative stress [18,19], excitotoxicity [20,21,22], and apoptosis [19,20,23,24]. This signalling is also implicated in cholesterol metabolism: it increases cholesterol production in neurons [25,26] and is regulated by cholesterol [27,28]. The role of TrkB signalling modulators in ferroptotic stress has not been studied to our knowledge; we, therefore, investigated the effects of several modulators of TrkB signalling (listed in Table 1) on sensitivity toward ferroptotic stress in cell culture.

## 2. Results

We first screened different immortalised cell lines, such as N27, HT-22, HT-1080, and U-251, as well as primary cortical neurons for their TrkB expression and found that only mature primary cortical neurons express detectable levels of TrkB receptors (Figure 1a). Mature primary neurons are resistant to erastin toxicity, so we tested the following TrkB modulators against sensitivity to RSL3 in these cells: agonist BDNF, antagonist ANA-12, and tyrosine kinase inhibitors K252a and GNF-5837. The TrkB inhibitor, GNF-5837, abolished ferroptosis in mature primary cortical neurons, but other modulators had no effect (Figure 1b–e).

Independent studies have shown that the TrkB activators 7,8-DHF [29,30] and phenelzine are protective against neurotoxicity [31,32]. We tested 7,8-DHF and phenelzine to determine whether they influence ferroptosis in neuronal and non-neuronal cell lines such as N27, HT-1080, and U-251 cells, which all lack detectable TrkB receptor expression. Co-treatment with different doses of 7,8-DHF conferred protection against erastin- and RSL3-induced ferroptosis in all tested cell lines (Figure 2a–d). We used N27 cells to compare 7,8-DHF rescue with the reference anti-ferroptotic compounds Lip-1 (a radical-trapping antioxidant, RTA) and deferiprone (an iron chelator). We found that 7,8-DHF was more potent than deferiprone but less potent than Lip-1 (Figure 2a,b). Next, we assessed the effect of TrkB transactivator phenelzine on ferroptotic stress in N27 and HT-1080 cells. Like 7,8-DHF, we found that phenelzine protected against erastin- and RSL3-induced ferroptosis in both cell lines (Figure 2e,f), but it was more effective in rescuing HT-1080 cells (EC_50_ against RSL3: 19.2 μM) than in rescuing N27 cells (EC_50_ against RSL3: 72.1 μM).

Moreover, 7,8-DHF and phenelzine significantly inhibited toxicity induced by the other ferroptosis activators FINO2 and tBH (Figure 3a,b). Since these immortalised cell lines did not express detectable TrkB receptors (Figure 1a), we hypothesised that ferroptosis inhibition might instead be due to intrinsic RTA properties of these compounds. In a cell-free 2, 2’-azino-bis (3-ethylbenzthiazoline-6-sulphonic acid (ABTS) assay, we confirmed that both compounds have RTA activity (IC_50_ of 7,8-DHF: 24.3 μM; phenelzine: 40.9 μM), which was commensurate to the reference anti-ferroptotic RTA, Lip-1 (IC_50_ =22.7 μM) (Figure 3c).

As we found that TrkB inhibitor GNF-5837 prevented ferroptosis in mature primary cortical neurons with an effect similar to the reference anti-ferroptotic molecule, Lip-1 (Figure 1c), we further characterised its efficacy in other cell lines. Consistent with the mature primary cortical neuron results, we observed potent anti-ferroptotic rescue against erastin (EC_50_: 36~116 nM) and RSL3 (EC_50_: 29~73 nM), commensurate to Lip-1 in the following immortalised cell lines: N27, HT-22, and HT-1080 (Figure 4a–f).

We further assessed the toxicity of GNF-5837 and found that up to 5 μM was not toxic after prolonged exposure (48 h) in the same cell lines: N27, HT-22, and HT-1080 (Figure 5a). We also found that GNF-5837 significantly inhibits toxicity in the N27 cells caused by other ferroptosis inducers, including peroxide compounds such as FINO2, tBH, and ammonium iron (II) sulphate (Figure 5b). The anti-ferroptotic activity of GNF-5837 was confirmed with a lipid peroxidation (C11 BODIPY) assay. Erastin, BSO, and RSL3 show increased lipid peroxidation, which was markedly suppressed by treatment with either Lip-1 (500 nM) or GNF-5837 (500 nM) in the N27 cells (Figure 5c). GNF-5837 is a coloured compound incompatible with the ABTS assay readout, so we assayed the compound’s action on generating the lipid peroxidation product, malondialdehyde (MDA), in a cell-free system. We found that GNF-5837 quenched the generation of MDA in this assay, with a similar potency to Lip-1 as a reference compound. Since GNF-5837 is effective in cells that do not express detectable levels of TrkB receptor (Figure 1a), and this compound quenches lipid radicals in a cell-free system (Figure 5d), our findings strongly suggest that the anti-ferroptotic activity of GNF-5837 is attributable to an RTA property, and not to receptor-mediated effects.

## 3. Discussion

Growing evidence of altered BDNF-TrkB signalling in neurodegeneration has spurred interest in understanding the possible underlying mechanisms. Ferroptosis is also implicated in neurodegeneration, and TrkB trophic signalling conceivably interferes with this pathway. Here, we reveal that several modulators of TrkB inhibit ferroptosis via a receptor-independent RTA mechanism.

While BDNF is a potent agonist of the TrkB receptor, systemic therapy with BDNF has several pharmacokinetic limitations, such as short half-life, large molecular size, and poor blood–brain barrier (BBB) penetration. As an orally bioactive flavonoid, 7,8-DHF penetrates the BBB to act as an agonist of the TrkB receptor, and numerous preclinical studies [29,30] demonstrate that this compound exhibits BDNF mimetic activity. The 7,8-DHF flavonoid ameliorated memory impairment and neuronal cell death in several studies of AD mouse models [29,30,33]. While 7,8-DHF has been reported as a TrkB agonist, it was unclear whether neuroprotection depends on the activation of the TrkB signalling pathway. We found that 7,8-DHF possesses potent anti-ferroptotic activity in cell lines that do not express detectable TrkB receptors, suggesting its anti-ferroptotic activity was due to another mechanism, such as the RTA activity that we showed in a cell-free assay. Therefore, it is possible that neuroprotection by 7,8-DHF was due to its RTA property in those prior studies rather than to TrkB signalling.

Typical and fast-acting antidepressants bind to the TrkB receptor to exert biological and behavioural actions [26,34]. Antidepressants such as phenelzine transactivate the TrkB receptor independently of BDNF and monoamine transporter blockade in mouse brains [35]. Like 7,8-DHF, we identified the TrkB-independent anti-ferroptotic activity of phenelzine against diverse ferroptosis inducers in various cell lines. Several in vitro and in vivo studies reported that the neuroprotective effect of phenelzine was via ameliorating oxidative damage caused by reactive oxygen species and reactive nitrogen species [31,32,36]. Phenelzine can chemically react with several reactive aldehydes, such as acrolein and 4-hydroxy-2-nonenal, due to its hydrazine-containing structure. So, the anti-ferroptotic activity of phenelzine that we observed in cell culture is most consistent with its RTA effects, as evidenced in the current report by its ability to scavenge ABTS radicals.

ANA-12 is a selective and potent antagonist of the TrkB receptor that penetrates the BBB [37], and K252a is a potent tyrosine kinase inhibitor that blocks neurotrophin signalling by Trk receptors [24,38,39]. Antagonism or inhibition of the TrkB receptor could lead to neuronal cell death; however, a low dose of systemic administration of ANA-12 surprisingly downregulates TrkB activity without affecting neuronal survival and demonstrates anxiolytic and antidepressant properties in mice [37]. K252a can prevent TrkB signalling, but it also shows neuroprotective activity via the prevention of apoptotic cell death in several preclinical studies [40,41,42]. While these compounds modulate TrkB signalling, we could not show their modulatory effects on ferroptotic stress in mature primary cortical neurons.

GNF-5837 is a potent and orally bioavailable oxindole pan-Trk inhibitor with a mechanism similar to K252a [43]. In our study, GNF-5837 showed anti-ferroptotic activity similar to that of liproxstatin 1 in various cell lines, either with or without TrkB receptor expression. Several oxindole compounds have shown protection against oxytosis and ferroptosis in cell culture studies [44,45,46]. Therefore, the anti-ferroptotic activity of GNF-5837 might be related to its oxindole RTA chemistry, in agreement with our cell-free assay that prevents MDA generation.

## 4. Materials and Methods

### 4.1. Drugs and Chemicals

RPMI 1640 culture media (Cat# 72400120), penicillin and streptomycin (Cat#15140122), B27 (Cat#17504044), and MTT (Cat#M6494) were purchased from Thermo Fisher Scientific (Waltham, MA, USA); Foetal Bovine Serum (FBS; Bovogen, Keilor East, Australia); Erastin (Cat#S7242), RSL3 (Cat#S8155), and 7,8-DHF (Cat#S8319) were purchased from Selleckchem (Selleck Chemicals LLC, Houston, TX, USA); FINO2 (Cayman Chemical, Ann Arbor, Michigan, USA); tBH solution (Cat#416665), BSO (Cat#B2515), deferiprone (Cat#379409), Lip-1 (Cat#SML1414), K252a (Cat#05288), recombinant human BDNF protein (Cat#B3795), phenelzine sulphate salt (Cat#P6777), GNF-5837 (Cat#SML0844), and ANA-12 (Cat#SML0209) were purchased from Sigma-Aldrich (Sigma-Aldrich Pty Ltd, an affiliate of Merck KGaA, Darmstadt, Germany).

### 4.2. Cell Culture

N27 rat dopaminergic neural cell line harvested from E12 rat mesencephalic tissue and HT-22 mouse hippocampal cell line subcloned from the HT-4 cell line (Merck, an affliate of Merck KGaA, Darmstadt, Germany), U-251 human glioblastoma—a kind gift from Associate Professors Andrew Morokoff and Kate Drummond—(The University of Melbourne, Melbourne, VIC, Australia), and HT-1080 human fibrosarcoma cells (CellBank Australia, Westmead, NSW, Australia) were cultured in RPMI 1640 media (supplemented with 10% FBS and 1% penicillin and streptomycin). As previously described, primary cortical neurons were isolated from E14 C57BL/6J mice (RRID: IMSR_JAX:000664) [47]. Upon isolation, primary cortical neurons were maintained in neurobasal media supplemented with 2% B27 (contains numerous antioxidant components, including alpha-tocopherol, vitamin A, and sodium selenite), GlutaMAX, penicillin, and streptomycin. RSL3 does not cause ferroptosis in these cells in the presence of B27, so after 3 days in vitro (DIV), the cells were transferred to RPMI 1640 media supplemented with 10% FBS and 1% penicillin and streptomycin, with cytosine arabinoside (2 μM) to minimise astrocyte growth. Cells were used for experiments between DIV9 and DIV11. All cell-based assays were conducted in RPMI 1640 media supplemented with 10% FBS and 1% penicillin and streptomycin. All cells were cultured at 37 °C with 5% CO_2_.

### 4.3. Cell Viability Assay

The colourimetric MTT assay is routinely used to measure cell viability in the context of ferroptosis [10,11,13]. The assay principle is that nicotinamide adenine dinucleotide phosphate (NADPH)-dependent cellular oxidoreductase enzymes and/or succinate dehydrogenase in the mitochondria of metabolically active cells reduce the yellow MTT to purple formazan crystals. As such, metabolic changes may impact the MTT signal in the absence of cell death; however, in the context of acute ferroptosis, we have regularly observed the MTT results to parallel other viability assays, such that any metabolic contribution would be marginal. To conduct this experiment, we seeded immortalised cells in 96-well plates at a density of 2 × 10^4^ cells/well, while primary cortical neurons were 1 × 10^5^ cells/well. Cells were then co-treated with tested compounds and ferroptosis inducers for 24 h (but tBH was for 4 h), and 20  μL of MTT (5  mg/mL) was added to each well and incubated for 2–4 h. The overlying media were aspirated from each well, and DMSO was added to the wells to dissolve formed formazan crystals in viable cells. The absorbance at 570 nm of the dissolved formazan demonstrated viable cells and was measured by a PowerWave XS microplate spectrophotometer (BioTek Instruments, Winooski, VT, USA), and cell viability was expressed as a percentage of control cells.

### 4.4. Radical-Trapping Antioxidant Assay

ABTS radical scavenging assay was used to evaluate the possible RTA properties of tested compounds in cell-free conditions. The assay was done according to the described protocol by Lee et al. (2015) [48] with the following modification. First, the ABTS reagent was prepared by mixing ABTS (7  mM) with potassium persulfate (2.45  mM) at a 1:1 volumetric ratio, which was then allowed to generate free radicals (kept in the dark at room temperature for 16 h). Then, the solution was diluted with water to achieve absorbance between 0.4 and 0.6) at 734  nm. In the assay, 100  μL of the tested sample or vehicle was mixed with 100  μL of ABTS reagent in a 96-well microplate, and the mixture was then incubated at room temperature for 7–15  min. Upon incubation, the absorbance of the mixture was measured by a PowerWave XS microplate spectrophotometer (BioTek Instruments, Winooski, USA) at 734  nm. The ABTS radical scavenging property was measured using the following formula:ABTS radical scavenging (%) = [(Control absorbance−Sample absorbance)/Control absorbance] × 100

### 4.5. Lipid Peroxidation Measurement

C11 BODIPY (581/591), a fluorophore that responds to lipid peroxidation with an increase in fluorescence emitted in the green range and a parallel decrease in fluorescence emitted in the red range, was employed to quantify lipid peroxides. After seeding cells per the viability assay, cells were co-treated with test compounds (prepared in media containing C11 BODIPY at a final concentration of 2.5 μM) for 24 h. Cells were washed once with PBS and then supplemented with PBS before measuring fluorescence at 565/600 nm (excitation/emission) for red fluorescence and 477/525 nm for green fluorescence using a CLARIOstar microplate reader (BMG LABTECH, Mornington, VIC, Australia). Lipid peroxidation was presented as the ratio of green to red C11 BODIPY fluorescence [11].

We evaluated the possible RTA property of GNF-5837 in cell-free conditions by a fluorescence-based lipid peroxidation assay using the MDA Assay Kit (ab118970, Abcam) as previously reported [11]. First, the tested compound GNF-5837 and positive control Lip-1 were added to the plate before generating MDA by a reaction between arachidonic acid (100 μM) and ammonium iron (II) sulphate (20 μM). Thiobarbituric acid (TBA) was then added to the plate and incubated at 90 °C for 1 h to generate MDA-TBA adducts. Following incubation, the effects of the tested compound and positive control on MDA-TBA adducts were measured with relative fluorescence units at Ex/Em = 532/553 nm using a CLARIOstar microplate reader (BMG LABTECH, Mornington, VIC, Australia).

### 4.6. Western Blotting Analysis

Total protein was extracted from the cells by a RIPA lysis buffer containing 50 mM Tris, 150 mM NaCl, 0.1% SDS, 0.5% sodium deoxycholate, 1% Triton X-100 and cOmplete™, and an EDTA-free protease inhibitor cocktail (Roche; Cat#05056489001). Protein was separated by sodium dodecyl sulphate-polyacrylamide gel electrophoresis, which was then transferred to polyvinylidene difluoride membranes using the iBlot Gel Transfer Device (Invitrogen). The membranes were blocked in tris-buffered saline containing 0.1% Tween-20 (TBST) and additionally containing 5% (*w/v*) skimmed milk for 1 h at room temperature and then incubated overnight at 4 °C with primary antibodies against TrkB (80E3; Cell Signalling; 1;1000) and beta-actin (A5441; Sigma; 1;5000), and then following incubation with the normal isotype-matched IgG-HRP conjugated antibodies (Dako) with a dilution ratio of 1:2000 for 1 h. Finally, immunoblots were visualised by a chemiluminescence reagent (Amersham ECL Start Western Blotting Detection Reagent) and imaged using a Fujifilm Luminescent Image Analyser LAS-3000 and then quantified by Image J software.

### 4.7. Statistical Analysis

Microsoft Excel v16 and GraphPad Prism 9.1 software for windows (GraphPad Software, San Diego, CA, USA, www.graphpad.com) were used to perform statistical analysis. Some test compounds were coloured; the operator and data assessment were not blinded, and samples were not randomised. At least 2–3 technical replicates were used to ensure the dependability of single values within each independent experiment.

Cell viability, cytotoxicity, and lipid peroxidation data were normalised to control samples. Results are shown as means ± SEM from at least 6–9 replicates from 2–3 independent experiments.

Non-linear regression analysis with a variable slope model (four parameters) was employed to fit a logistic curve to dose–response data to determine EC_50_ and IC_50_ with 95% CI using Prism (Graphpad). The corresponding fitted regression curve is shown on the data where an EC_50_ or IC_50_ is reported. One- or two-way ANOVA was used for all multiple comparisons to calculate statistical significance, and *p* < 0.05 was deemed statistically significant.

## 5. Conclusions

We found that several potent modulators of TrkB signalling, such as BDNF, K252a, and ANA-12, did not affect ferroptotic lethality in TrkB-expressing mature primary cortical neurons. BDNF mimetic 7,8-DHF, TrkB activator phenelzine, and inhibitor GNF-5837 showed TrkB-independent anti-ferroptotic activity, possibly acting as RTAs. Caution should be applied when interpreting the results of these compounds in prior and future studies since the potent RTA effect may confound the result. As we detected TrkB receptor expression only in mature cortical neurons, this study was limited to characterising the effect of TrkB signalling on RSL3-induced ferroptotic lethality. We do not exclude the possibility that TrkB modulators affect ferroptotic stress caused by cystine/glutamate transporter inhibitors such as erastin in TrkB-expressing cell lines.

## Figures and Tables

**Figure 1 ijms-23-16205-f001:**
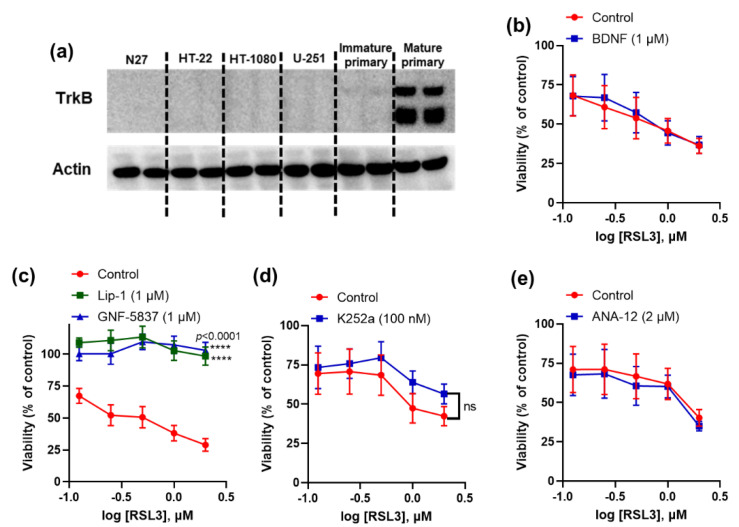
Activity of TrkB modulators against ferroptosis in mature primary cortical neurons. Immunoblot shows TrkB expression in mature primary cortical neurons (**a**). GNF-5837 inhibits (while other TrkB modulators do not affect) RSL3-induced ferroptosis in primary cortical neurons (**b**–**e**). Cells were co-treated with ferroptosis inducer RSL3 and tested compounds for 24 h. Cell viability was measured by 3-(4,5-dimethylthiazol-2-yl)-2,5-diphenyltetrazolium bromide (MTT) assay. Data points signify mean percentage survival relative to untreated controls ± SEM, *n* =  9–10 from 2–3 independent experiments. *p* values were calculated using the two-way ANOVA (Dunnett’s multiple comparisons test, (**c**) and the unpaired t-test (**d**), and the significant level is noted as **** *p* < 0.0001 (vs. control).

**Figure 2 ijms-23-16205-f002:**
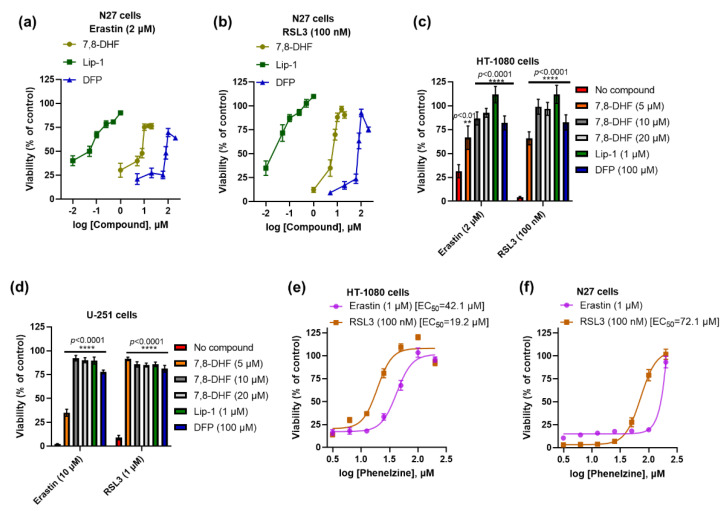
Both 7,8-DHF and phenelzine protect ferroptosis in several immortalised cell lines. Efficacy of 7,8-DHF, Lip-1, and DFP against erastin- and RSL3-induced ferroptosis in N27 cells (**a**,**b**). Like Lip-1 and DFP, 7,8-DHF significantly prevents ferroptosis in HT-1080 and U-251 cells (**c**,**d**). Efficacy of phenelzine against erastin- and RSL3-induced ferroptosis in HT-1080 and N27 cells (**e**,**f**). Cells were co-treated with ferroptosis inducers and tested compounds for 24 h. Cell viability was measured by MTT assay. Data points signify mean percentage survival relative to untreated controls ± SEM, *n* =  12–20 from 3–5 independent experiments. *p* values were calculated using the two-way ANOVA (Šídák’s multiple comparisons test), and the significant levels are noted as ** *p* < 0.01 and **** *p* < 0.0001 (vs. no compound).

**Figure 3 ijms-23-16205-f003:**
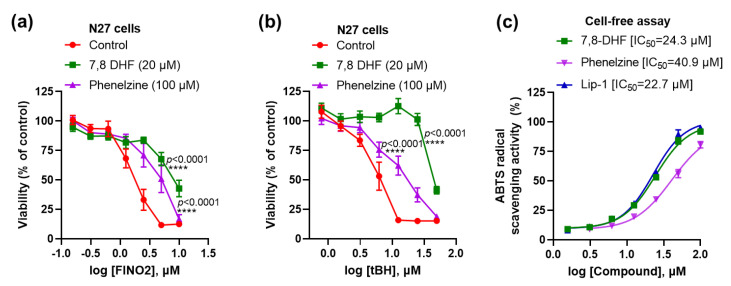
Both 7,8-DHF and phenelzine protect against ferroptosis and have RTA properties. Both 7,8-DHF and phenelzine attenuate FINO2- and tBH-induced toxicity (**a**,**b**). RTA properties of 7,8-DHF, phenelzine and Lip-1 in cell-free ABTS assay (**c**). Cells were co-treated with ferroptosis inducers and tested compounds for 24 h, except for tBH, which had a 4 h co-treatment. Cell viability assay was measured by MTT assay. Data points signify mean percentage survival relative to untreated controls ± SEM, *n* =  9–16 from 3–4 independent experiments. P values were calculated using the two-way ANOVA (Dunnett’s multiple comparisons test), and the significant level is noted as **** *p* < 0.0001 (vs. control).

**Figure 4 ijms-23-16205-f004:**
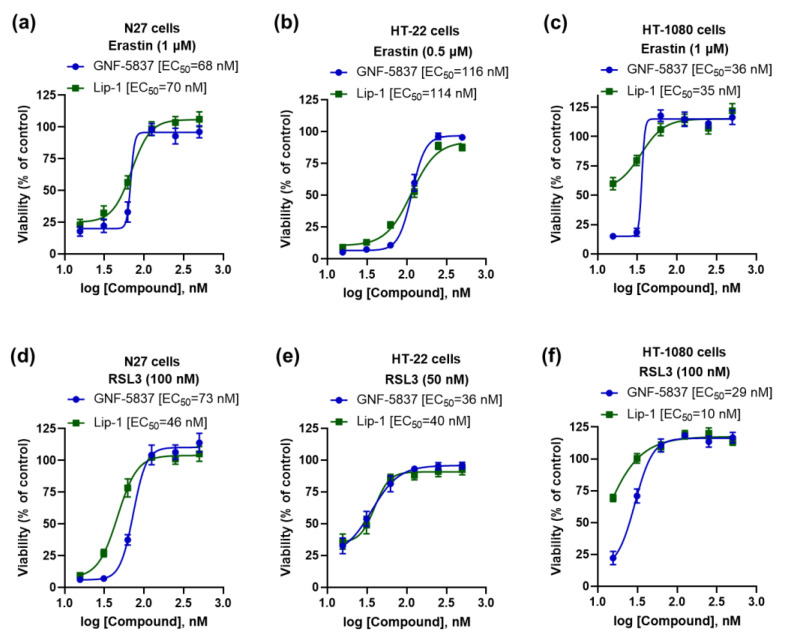
GNF-5837 protects ferroptosis in several immortalised cell lines. Comparative anti-ferroptotic activity between GNF-5837 and Lip-1 for erastin- and RSL3-induced ferroptosis following 24 h incubation in different cell lines: N27, HT-22, and HT-1080 cells (**a**–**f**). Cell viability was measured by MTT assay. Data points signify mean percentage survival relative to untreated controls ± SEM, *n* = 11–16 from 3–4 independent experiments.

**Figure 5 ijms-23-16205-f005:**
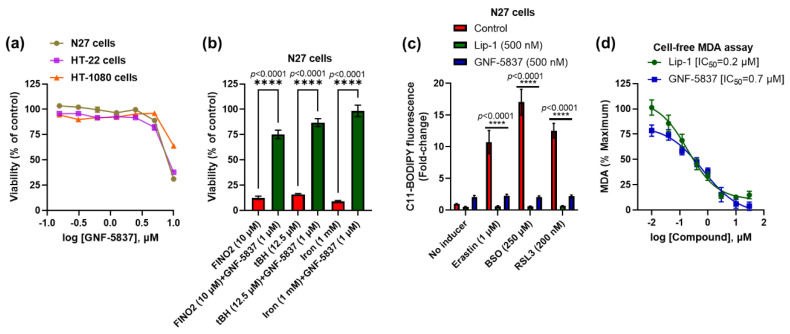
GNF-5837 protects against ferroptosis likely via its RTA property. Toxicity assay upon 48 h treatment with GNF-5837 in different cell lines (**a**). GNF-5837 protects against toxicity induced by FINO2, tBH, and iron in N27 cells (**b**). Like Lip-1, GNF-5837 significantly protects against lipid peroxidation in the C11 BODIPY assay caused by erastin, BSO, and RSL3 in N27 cells (**c**). RTA property of GNF-5837 and Lip-1 against lipid peroxidation in cell-free MDA assay (**d**). Cells were co-treated with ferroptosis inducers and tested compounds for 24 h, except for tBH-induced ferroptosis, which had a 4 h co-treatment, and MTT assay measured cell viability. Lipid peroxidation in C11 BODIPY assay is expressed as the ratio of green to red C11 BODIPY fluorescence. Data points signify mean percentage survival relative to respective controls ± SEM (*n* = 36 for (**a**), *n* =  9–12 for (**b**,**c**), and *n* = 6 for (**d**) from 3–4 independent experiments). P values were calculated using the one-way ANOVA (Tukey’s multiple comparisons test for (**b**)) and two-way ANOVA (Dunnett’s multiple comparisons test for (**c**)), and the significant level is noted as **** *p* < 0.0001 (vs. inducers/control).

**Table 1 ijms-23-16205-t001:** Modulators of TrkB receptor signalling tested in this study.

Compound	Character
BDNF	Selective TrkB agonist
7, 8-dihydroxyflavone (7,8-DHF, a flavonoid)	Non-selective TrkB agonist
Phenelzine sulphate (antidepressant; monoamine oxidase inhibitor)	Non-selective TrkB activator
ANA-12 (a synthetic agent)	Selective, non-competitive TrkB antagonist
K252a (an alkaloidal compound isolated from *Nocardiopsis* bacteria)	Tyrosine kinase inhibitor
GNF-5837 (a synthetic oxindole compound)	Tyrosine kinase inhibitor

## Data Availability

Not applicable.

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
