# Peer review of "Receptor-Independent Anti-Ferroptotic Activity of TrkB Modulators"

_ijms, 2022, doi:10.3390/ijms232416205_

Round 1

Reviewer 1 Report

This  manuscript  addresses  TrkB signalling and  ferroptosis. The manuscript  should focus  on cells  that respond, the  others  could  be just mentioned.

1)     Not  sure what the  purpose  of  the  chemical structure si ?

2)     There should be a  section addressing  the MTT assay especially as  oxidative  stress is  involved.

3)     Line 294   please provide more  details  on which Roche  product

4)     It is not clear how  the the  dose-response  curves  are reported. Graphpad calculations are referred  to, yet  the  confidence  intervals and R2 are  not reported. Furthermore  many curves  appare  to be  more a point-to point  drawing rather than a curve as  such.

5)     By focusing  on responsive cells, and  perhabs a summary table  the manuscript  can  be  improved.

Author Response

Comments and Suggestions for Authors

Reviewer 1:

This  manuscript  addresses  TrkB signalling and  ferroptosis. The manuscript  should focus  on cells  that respond, the  others  could  be just mentioned.

Response: A major TrkB-interacting compound GNF-5837 that we demonstrate to have potent anti-ferroptotic activity in this manuscript was effective in all the cell types that we investigated (primary neurons, N27, HT-1080, HT22). We believe it is important to show the results in different cells that do not express detectable TrkB (N27, HT-1080, HT22) to demonstrate that anti-ferroptotic action of GNF-5837 does not occur via interactions with this receptor.  

1)     Not  sure what the  purpose  of  the  chemical structure si ?

Response: We appreciate the reviewer's concern about chemical structure. Since we did not discuss the chemical structure of all compounds, we have removed the chemical structure from Table 1.

2)     There should be a  section addressing  the MTT assay especially as  oxidative  stress is  involved.

Response: We have added the following text: "The colourimetric MTT assay is routinely used to measure cell viability in the context of ferroptosis [1-3]. The assay principle is nicotinamide adenine dinucleotide phosphate (NADPH)-dependent cellular oxidoreductase enzymes and/or succinate dehydrogenase in the mitochondria of metabolically active cells reduce the yellow MTT to purple formazan crystals. As such, metabolic changes may impact the MTT signal in the absence of cell death, but in the context of acute ferroptosis, we have regularly observed the MTT results to parallel other viability assays, such that any metabolic contribution would be marginal".

3)     Line 294   please provide more  details  on which Roche  product

Response: We have provided more details on Roche product: "cOmplete™, EDTA-free protease inhibitor cocktail protease inhibitors (Roche; Cat#05056489001)"

4)     It is not clear how  the the  dose-response  curves  are reported. Graphpad calculations are referred  to, yet  the  confidence  intervals and R2 are  not reported. Furthermore  many curves  appare  to be  more a point-to point  drawing rather than a curve as  such.

Response: We appreciate the reviewer’s concern about data presentation. We have revised the statistical analysis and figures as follows:  Non-linear regression analysis with a variable slope model (four parameters) was employed to fit a logistic curve to dose-response data to determine EC50 and IC50 with 95% CI using Prism (Graphpad). The corresponding fitted regression curve is shown on the data where an EC50 or IC50 is reported. One and two-way ANOVA was used for all multiple comparisons to calculate statistical significance and p<0.05 was deemed statistically significant’’.  

5)     By focusing  on responsive cells, and  perhabs a summary table  the manuscript  can  be  improved.

Response: TrkB modulators showed a similar effect in all cell types tested; therefore, we did not provide an additional summary table.

Reviewer 2 Report

NIL

Author Response

Reviewer 2:

Jakaria Md et al, in the study entitled "Receptor-independent anti-ferroptotic activity of TrkB modulators" has used several modulators of TrkB that is involved in several neuro degenerative diseases including Alzheimer's.

Comments:

The work performed is well appreciated. However, a few minor corrections need to be addressed viz…

Replace the word 'such as' with 'Including' in line No. 10 and in line no. 38 of Page 1

Response: We have replaced it based on the reviewer's suggestion.

Description and details of B27 is needed in line 239 of page 8.

Response: We have explained B27 in the revised manuscript: "B27 (Cat#17504044)" in line no 221 and "2% B27 (contains numerous antioxidant components, including alpha-tocopherol, vitamin A and sodium selenite), GlutaMAX, penicillin and streptomycin. RSL3 does not cause ferroptosis in these cells in ---" no 233-234.

The authors could have included the estimation of iron content in the test cells along with the estimation of lipid peroxides since iron is equally involved in the process of ferroptosis.

Response:   It is unlikely that the major reported novel compound GNF-5837 (~120 nM EC50) impacts iron, for example by chelation, because its effective concentration against ferroptosis is much lower than bona fide iron chelators (50-100μM). This is especially evident in the cell free assay, where the potency of this compound is less (~0.7uM), which is typical of anti-ferroptotic compounds such as liproxstatin, but the iron levels in this assay well exceed this concentration (10 uM). These stoichiometry comparisons demonstrate that the compound doesn’t function by chelating iron. Regarding measurement of lipid peroxides – please refer to the C11-BODIPY data presented in Fig 5C.   

Only RTA or antioxidant assay cannot conclude the activation of ferroptosis, in order to prove the anti ferroptotic property more specific assays, may be western blot images of ferroptotic molecules has to be included.

Response: We activated ferroptosis using a range of well-established ferroptosis induces, and we demonstrated that cell death is abolished by a bona fide ferroptosis inhibitor, liproxstatin. Therefore, ferroptosis has been established. We are not suggesting that any of the compounds that we profile activate ferroptosis. We do not know of a way to prove ferroptosis inhibition by western blot measurement of proteins – rather, ferroptosis inhibition is demonstrated by response against a range of established ferroptosis inducers, which we done. Measurement of proteins may give insight into biological mechanisms, but we demonstrate the activity of these drugs in a cell free system, so their activities do not depend on a cellular response.

The author may justify whether highlighting the TrkB is appropriate when most of the undergone mechanism suggested TrkB independent ferroptosis.

Response: The compounds we tested are well-known for their potent TrkB modulatory role.  Since these compounds belong to the same class, "TrkB", we have highlighted TrkB, although we have shown that their anti-ferroptotic effects were independent of the TrkB signalling. Since these compounds are regularly used as BDNF mimetics to infer biological activity regarding the BDNF/TrkB system, we believe it is important to communicate to the field this important off-target effect.

Round 2

Reviewer 2 Report

NIL